# GaussGym:
# An open-source real-to-sim framework for learning locomotion from pixels

## Abstract

We present a photorealistic robot simulator that integrates 3D Gaussian Splatting as a drop-in renderer within vectorized physics simulators such as IsaacGym. This enables unprecedented speed—exceeding 100,000 steps per second on consumer GPUs—while maintaining high visual fidelity, which we showcase across diverse tasks. We additionally demonstrate its applicability in a sim-to-real robotics setting. Beyond depth-based sensing, our results highlight how rich visual semantics improve navigation and decision-making, such as avoiding undesirable regions. We further showcase the ease of incorporating thousands of environments from iPhone scans, large-scale scene datasets (e.g., GrandTour, ARKit), and outputs from generative video models like Veo, enabling rapid creation of realistic training worlds. This work bridges high-throughput simulation and high-fidelity perception, advancing scalable and generalizable robot learning, and allowing researchers to benchmark their visual locomotion algorithms. All code and data will be open-sourced for the community to build upon. Videos, code, and data are available on the project website: `https://gauss-gym.com`.

## 1 Introduction

For mobile robots to act in unstructured real-world settings, they need to be able to accurately perceive the environment around them (Gervet et al., 2023; Chang et al., 2023). Consider a robot that needs to reach target locations within the environment while navigating obstacles and interacting with man-made objects. Many such obstacles and environment affordances are only detectable through visual observations, such as crosswalks, puddles, or colored features.

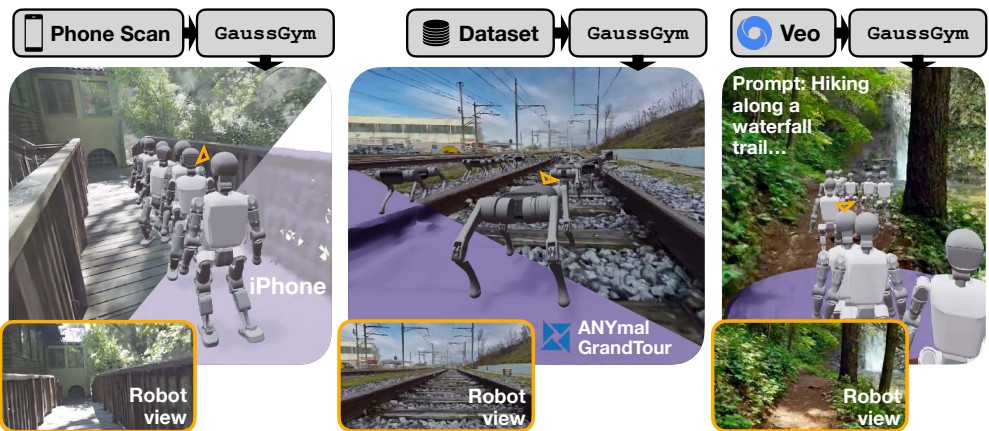

Figure 1: GaussGym constructs photorealistic worlds from various data sources and renders them in a vectorized physics engine, achieving high visual fidelity and throughput.

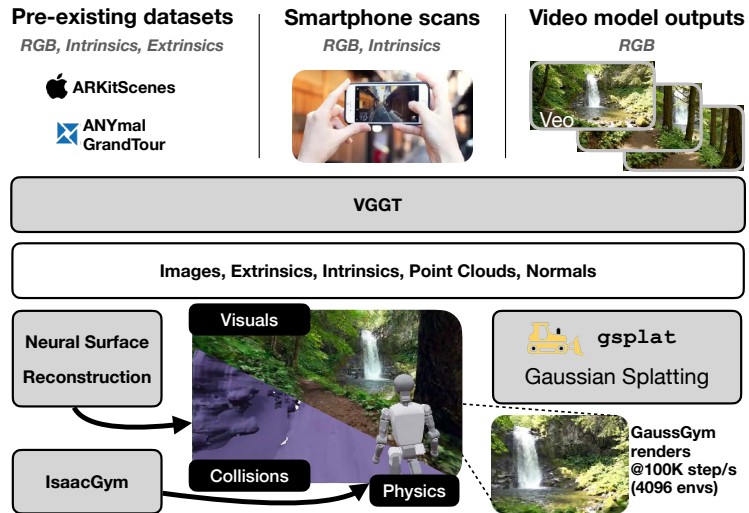

Figure 2: *Data collection overview*: GaussGym ingests data from various data sources and processes them with VGGT (Wang et al., 2025) to obtain extrinsics, intrinsics, and point clouds with normals. The former two data products are used to train 3D Gaussian Splats for rendering, while the latter two are used to estimate the scene collision mesh.

The dominant paradigm for achieving locomotion on legged robots, sim-to-real (Hwangbo et al., 2019) reinforcement learning (RL), faces considerable challenges in fully leveraging visual properties of real-world environments. In principle, this approach allows a control policy trained in simulation to transfer to a real robot without adaptation, achieving robust locomotion. While existing simulators (Makoviychuk et al., 2021a; Todorov et al., 2012; Tao et al., 2024; Genesis, 2024) capture physics with sufficient fidelity for transfer, their treatment of visual information is often either too slow or too inaccurate, limiting the effectiveness of policy learning and transfer. Consequently, most perceptive locomotion frameworks in the literature rely on LiDAR or depth inputs (Hoeller et al., 2024), which restrict policies from exploiting semantic cues in the environment and narrow the range of tasks that can be realistically pursued in simulation.

With GaussGym, we present an open-source simulation framework that digitizes real-world and video model–generated environments, and simulates both their physics and photorealistic renderings to enable learning locomotion and navigation policies directly from RGB pixels. GaussGym builds on advances in 3D reconstruction and differentiable rendering to bring diverse input sources into simulation. The system is designed to accept a wide range of data, including smartphone scans, fully sensorized SLAM captures, existing 3D datasets, hand-held videos, and even outputs from generative video models. GaussGym is highly efficient, simulating hundreds of thousands of environment steps per second across 4,096 robots at $640 \times 480$ resolution on a single RTX 4090 GPU. We also consider GaussGym an effective benchmark for testing visual locomotion strategies.

To demonstrate the effectiveness of GaussGym for training visuomotor policies with RL, we train locomotion and navigation policies for both humanoid and quadrupedal robots. Despite the increased throughput and visual fidelity of GaussGym, training directly from RGB remains challenging, as policies must infer geometry from vision rather than rely on provided heightmaps or depth images. We address this by incorporating an auxiliary reconstruction loss guided by ground-truth mesh data, which significantly improves learning speed and performance. Finally, we show initial zero-shot transfer of visual locomotion policies trained in GaussGym to real-world stair climbing, marking a first step toward closing the visual sim-to-real gap. Beyond this demonstration, GaussGym democratizes access to photorealistic simulation and lays the foundation for future research on visual locomotion and navigation.

We summarize our contributions below:

1. GaussGym: a fast open-source photorealistic simulator with 5,000 scenes, supporting diverse scene creation from manual scans, open-source datasets, and generative video models.
2. We share findings on addressing the visual sim-to-real gap, showing that incorporating geometry reconstruction as an auxiliary task significantly improves stair-climbing performance.

3. We demonstrate the semantic reasoning of RGB navigation policies in a goal-reaching task, where policies trained on pixels successfully avoid undesired regions that are invisible to depth-only policies.

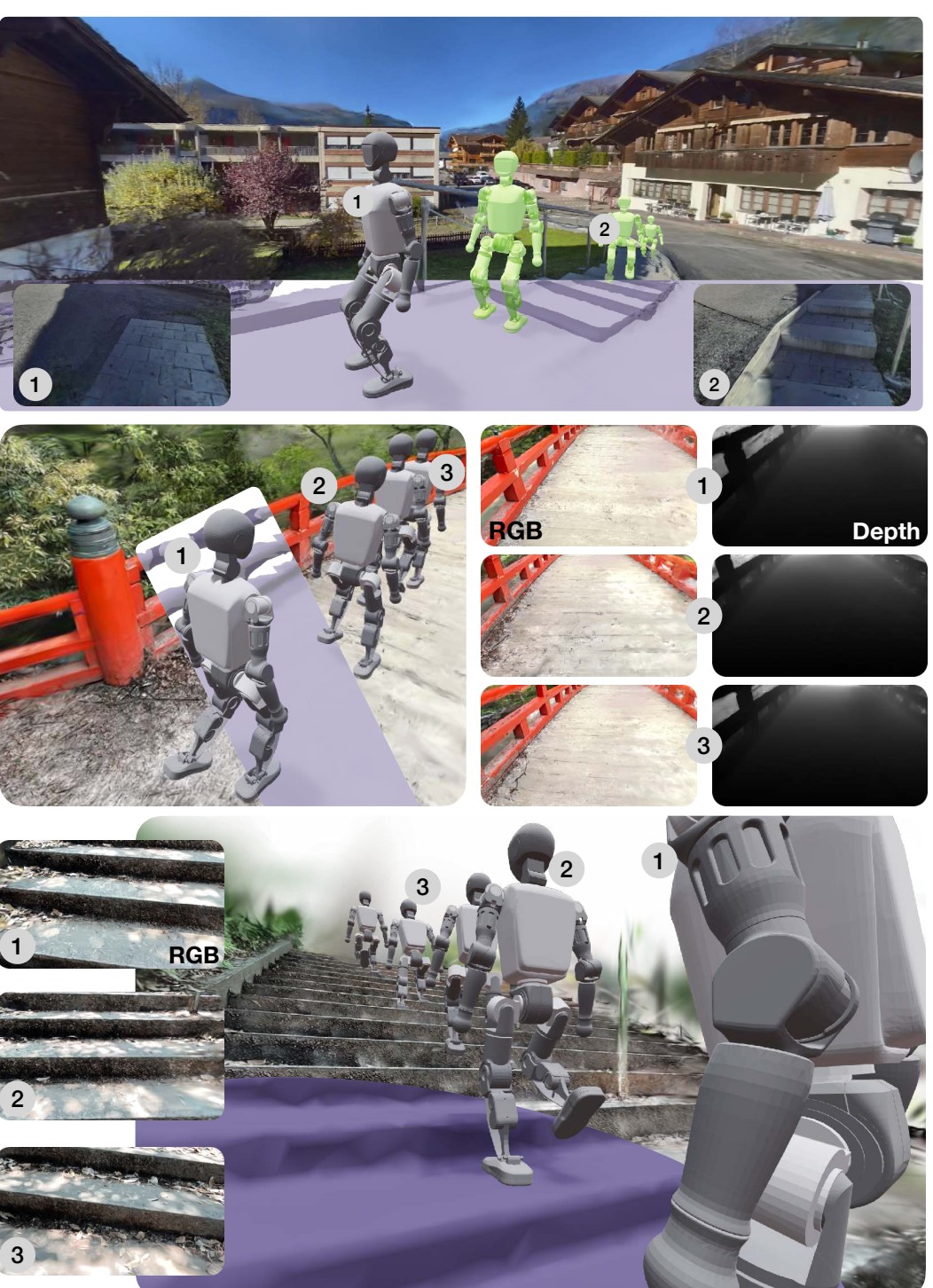

Figure 3: *Velocity-tracking policies trained directly from pixels in GaussGym*: Photorealistic environments provide diverse training scenes, enabling policies to follow commanded velocities from RGB input. GaussGym supports large-scale training with synchronized RGB and depth at over 100K $\frac{steps}{s}$ across 4,096 parallel environments on an RTX4090 rendering at $640 \times 480$.

## 2 RELATED WORK

### 2.1 SIM-TO-REAL RL FOR LOCOMOTION

Simulation provides a scalable and cost-effective method for training RL locomotion and navigation policies, avoiding costly hardware data collection and unsafe real-world exploration while granting access to privileged information during training. The ideal simulator for developing these policies comprises several key properties: high throughput, accurate physics, and photorealistic rendering.

While rigid-body-dynamics CPU-based simulators like MuJoCo (Todorov et al., 2012), PyBullet (Coumans & Bai, 2016–2021), and RaiSim (Hwangbo et al., 2018) enabled training and transferring of RL locomotion policies from simulation to the real world (Tan et al., 2018), the advent of GPU-accelerated simulators has democratized RL training by leveraging consumer-grade hardware for simulation. Platforms such as Isaac Gym (Makoviychuk et al., 2021a), Isaac Sim (Makoviychuk et al., 2021b), and others (Tao et al., 2024; Zakka et al., 2025; Genesis, 2024) have been instrumental in this progression, supporting the rapid development and advances in legged locomotion (Rudin et al., 2021) and navigation (Lee et al., 2024).

Despite frameworks such as IsaacLab (Makoviychuk et al., 2021b), ManiSkill (Tao et al., 2024), and Genesis (Genesis, 2024) supporting parallelized hardware-accelerated rendering, most locomotion policies deployed in the real world are restricted to geometric (e.g., depth, elevation maps) and proprioceptive inputs. This can be explained by the visual-sim-to-real gap, lack of diverse assets capturing the real world, and the high throughput required for training RL policies. Implicit learned scene representations, such as 3D Gaussian Splatting (3DGS) (Kerbl et al., 2023), offer a compelling alternative, directly improving visual fidelity and rendering throughput.

### 2.2 SCENE GENERATION

Heuristic and handcrafted rules (Rudin et al., 2021), as well as procedural terrain generation (Lee et al., 2024), are commonly used to create environments for training locomotion and navigation policies. While these heuristic-based rules are effective for defining geometric terrains that lead to robust locomotion behaviors, they do not allow for specifying a meaningful visual appearance of the scene. Achieving realistic visuals requires composing scenes from textured assets. Some works have attempted to import assets to be used for learning locomotion directly from video using SfM methods, however they do it without re-rendering the scene in RGB (Allshire et al., 2025). Asset libraries for realistic scene simulation are available through platforms like ReplicaCAD (Szot et al., 2021), LeVerb (Xue et al., 2025), and AI2-THOR (Kolve et al., 2017) (including iTHOR and RoboTHOR) or can be generated procedurally (Deitke et al., 2022). Alternatively, realistic scenes can be captured using specialized 3D scanners (Chang et al., 2017; Xia et al., 2018) and then further integrated into simulation frameworks like Habitat (Ramakrishnan et al., 2021). However, most rendering pipelines rely on textured-mesh assets, which often result in lower visual fidelity.

Our approach builds on NeRF2Real (Byravan et al., 2023), which improves visual fidelity by capturing scenes with a Neural Radiance Field (NeRF), followed by mesh extraction and manual post-processing to train a locomotion policy. However, it is computationally expensive due to slow ray-tracing and lacks vectorization support. (Zhu et al., 2025; Xie et al., 2024; Chhablani et al., 2025) construct 3D Gaussians of multiple environments and train a visual high-level navigation policy. Several works in robotic manipulation (Torne et al., 2024; Chen et al., 2024b) adopt similar strategies, using 3DGS to create articulated scenes or train models to predict an object's Unified Robot Description Format (URDF), including its actuation, from a single image (Chen et al., 2024b). LucidSim (Yu et al., 2024) makes two key contributions: first, it employs a ControlNet diffusion model to generate visual training data from depth maps and semantic masks; second, it introduces a real-to-sim framework by training 3DGS and manually aligning reference frames with meshes created using Polycam for a select set of test scenes. Today's state-of-the-art world and video models trained on internet-scale video data demonstrate unprecedented levels of controllable video generation (DeepMind, 2025; Bruce et al., 2024; Google DeepMind, 2025; Wan et al., 2025) and can synthesize multiple seconds of photorealistic, multi-view-consistent video. Although their slow inference speed renders them impractical as direct simulators, these models create opportunities to rethink scalable 3D asset and environment creation from simple text prompts. A comparison of simulators can be found in table 1.

### 2.3 Radiance Fields in Robotics

Neural Radiance Fields (NeRFs) (Mildenhall et al., 2020) are an attractive representation for high quality scene reconstruction from posed images, with an abundance of recent work on visual quality (Adamkiewicz et al., 2022; Barron et al., 2021; 2022; Ma et al., 2022; Huang et al., 2022; Sabour et al., 2023; Philip & Deschaintre, 2023), large-scale scenes (Tancik et al., 2023; Wang et al., 2023; Barron et al., 2023), optimization speed (Müller et al., 2022; Chen et al., 2022; Fridovich-Keil et al., 2023; Yu et al., 2021), dynamic scenes (Park et al., 2021; Li et al., 2023; Pumarola et al., 2020), and more. They have shown promise in robot manipulation, beginning with leveraging NeRF as a high-quality visual reconstruction for grasping (Kerr et al., 2022; Ichnowski* et al., 2020) and more recently by leveraging its ability to embed higher dimensional features for language-guided manipulation (Rashid et al., 2023; Shen et al., 2023). A core limitation of neural fields is their slow training speed, which 3D Gaussian Splatting (3DGS) mitigates (Kerbl et al., 2023) by representing radiance fields as a collection of oriented 3D gaussians which can be differentiably rasterized quickly on modern GPU hardware. Many works transfer high-dimensional feature fields to 3DGS for rapid training and rendering, as well as language-guided robot grasping, persistent Gaussian representations for manipulation, and visual imitation (Zheng et al., 2024; Qin et al., 2023; Qiu et al., 2024; Yu et al., 2025a;c; Kerr et al., 2024; Yu et al., 2025b).

Radiance Fields have also shown promise as large-scale scene representations for navigation as a differentiable collision representation (Adamkiewicz et al., 2022), as a visual simulator for learning drone flight or autonomous driving from RGB pixels (Khan et al., 2024; Chen et al., 2025), or as a scene representation to train locomotion affordance models with view augmentation (Escontrela et al., 2025). GaussGym draws inspiration from these results, but integrates high-fidelity environment visual simulation with contact physics from IsaacSim to enable locomotion. The most related prior work is LucidSim Yu et al. (2024), which develops a similar splat-integrated simulator for evaluating locomotion policies. GaussGym takes a similar real-to-sim approach, but implements a framework which easily scales to thousands of scanned scenes, integrates tightly with massively parallel physics simulation, and presents a flexible framework for future research to build on.

## 3 GaussGym

Figure 2 illustrates the overall GaussGym pipeline. Data can originate from posed datasets, casual smartphone scans, or even raw RGB sequences from video generation models. All inputs are standardized via the Visually Grounded Geometry Transformer (VGGT) (Wang et al., 2025), which estimates camera intrinsics, extrinsics, dense point clouds, and normals. These intermediate representations are then passed to a neural surface reconstruction module to generate meshes, while Gaussian splats are initialized directly from VGGT point clouds to provide accurate geometry and rapid convergence. The resulting assets are automatically aligned in a shared global frame. During simulation, Gaussian Splatting is used as a drop-in renderer, producing photorealistic visuals at

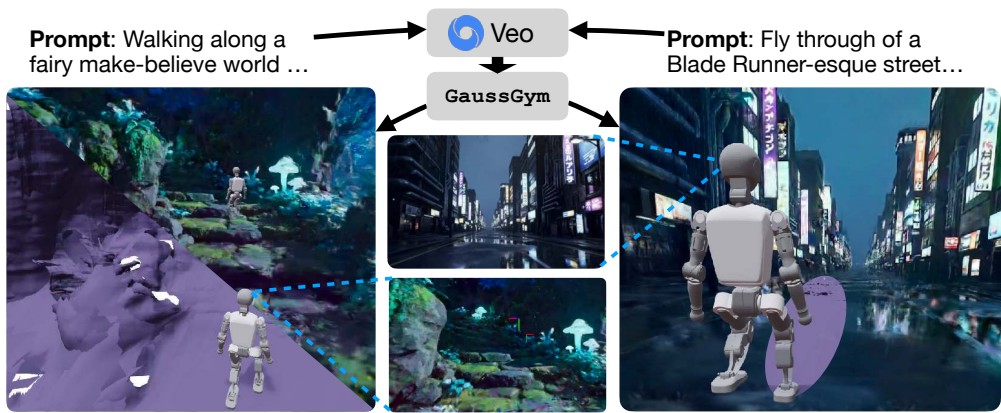

Figure 4: GaussGym ingests a variety of datasets - including video model outputs - to produce photorealistic training environments for robot learning.

| Method | GaussGym | LucidSim | LeVerb | IsaacLab |
|---|---|---|---|---|
| Photorealistic | ✓ | ✓ | ✗ | ✗ |
| Temporally consistent | ✓ | ✗ | ✓ | ✓ |
| FPS (vectorized) | 100,000[†] | Single env only | Not reported | 800[‡] |
| FPS (per env) | 25 | 3 | Not reported | 1 |
| Renderer | 3D Gaussian Splatting | ControlNet | Raytracing | Raytracing |
| Scene Creation | Smartphone scans, Pre-existing datasets, Video model outputs | Hand-designed scenes | Hand-designed scenes | Randomization over primitives |

Table 1: *Comparison of GaussGym to different simulators*: GaussGym and IsaacLab were configured to render at $640 \times 480$, LucidSim configured to render at $1280 \times 768$ [†]: Vectorized across 4096 envs on RTX4090. [‡]: Vectorized across 768 envs on RTX4090.

scale while remaining fully synchronized with physics for collision handling. This design allows GaussGym to combine diverse real-world and synthetic data sources with high-speed rendering for large-scale robot learning. Example scenes from various sources are visualized in fig. 1 and fig. 3.

## 3.1 DATA COLLECTION AND PROCESSING

GaussGym is designed to flexibly ingest data from a wide range of sources. These include posed datasets such as ARKitScenes (Baruch et al., 2021) and GrandTour (Frey et al., 2025), smartphone captures with intrinsic calibration, and even unposed RGB sequences generated by modern video models such as Veo (Google DeepMind, 2025).

All data are formatted into a common gravity-aligned reference frame before processing. We use VGGT to extract camera intrinsics, extrinsics, and dense scene representations including point clouds and surface normals. From these outputs, a Neural Kernel Surface Reconstruction (NKSR) (Huang et al., 2023) is used to produce high-quality meshes, while Gaussian splats are initialized directly from VGGT point clouds. Point-cloud initialization of Gaussian splats greatly improves geometric fidelity and accelerates convergence. Our approach achieves precise visual-geometric alignment, extending LucidSim's real-to-sim pipeline (Yu et al., 2024), which is limited to smartphone scans, requires manual registration of the mesh and 3DGS, and does not provide vectorized rendering.

## 3.2 3D GAUSSIAN SPLATTING AS A DROP-IN RENDERER

Once reconstructed, Gaussian splats are rasterized in parallel across simulated environments. Unlike traditional raytracing or rasterization pipelines (Xue et al., 2025; Makoviychuk et al., 2021a), splatting provides photorealistic rendering with minimal overhead and is highly amenable to vectorized execution. We batch-render splats across environments using multi-threaded PyTorch kernels, ensuring efficient GPU utilization and distributed training. Example RGB and depth renders for indoor and generative model scenes are are shown in fig. 5 and fig. 4.

## 3.3 OPTIMIZATIONS FOR HIGH-THROUGHPUT AND REALISM

To maximize efficiency, we decouple rendering from the proprioceptive control rate and simulation frequency: instead of rendering at the control frequency, we render at the camera's true frame rate, which is normally slower than the control frequency. This yields additional speed-ups while preserving high-fidelity visual input for the policy. To further reduce the Sim2Real gap, we introduce a simple but novel method to simulate motion blur: rendering a small set of frames offset along the camera's velocity direction and alpha-blending them into a single image, which produces realistic blur artifacts that improve visual fidelity and robustness in transfer. This is especially noticeable in scenes with sudden jolts, such as climbing stairs or high-speed movements. Example motion blur sequences are shown in Appendix fig. 10.

In practice, a single GPU can render up to 4,096 environments across 128 unique scenes at 100,000 simulator steps per second wall clock time, where the control and camera update rates in simulator

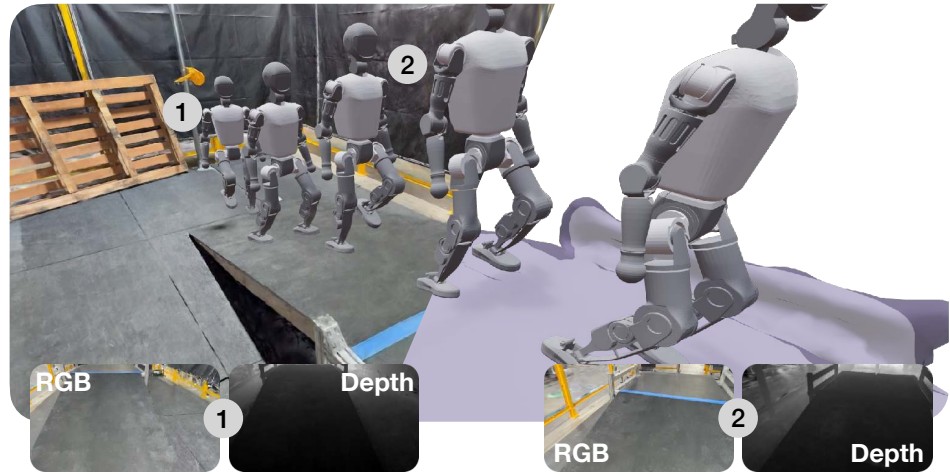

Figure 5: *Rendering RGB and Depth*: Since depth is a by-product of the Gaussian Splatting rasterization process, GaussGym also renders depth without increasing rendering time.

time are 50Hz and 10Hz, respectively (on an RTX 4090). Scaling is near-linear across multiple GPUs, enabling distributed training on thousands of diverse, photorealistic scenes simultaneously. This throughput makes it possible to train vision-based locomotion policies with a level of scene diversity and realism previously unattainable in high-speed simulators.

## 4 RESULTS

### 4.1 TRAINING ENVIRONMENTS BEYOND REALITY

GaussGym integrates data from smartphone scans and open-source datasets, but its standout capability is generating entirely new worlds from video models. This enables the creation of environments that are difficult or impossible to capture in the real world, such as caves, disaster zones, or even extraterrestrial terrains (Fig. 4). The key enablers are the strong multi-view consistency of Veo and the robust camera estimation and dense point cloud generation of VGGT. Additional scenes and videos are available on our webpage.

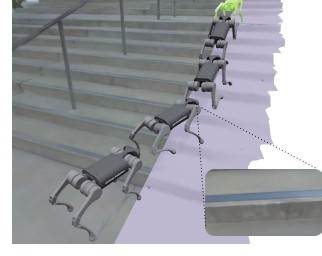

(a) RGB policy pre-trained in GaussGym.

### 4.2 VISUAL LOCOMOTION AND NAVIGATION

To evaluate the benefits of photorealistic rendering in GaussGym, we focus on the task of visual stair climbing and visual navigation in diverse visually complex terrains. We specifically choose to use an asymmetric actor-critic framework to learn from visual input, rather than relying on student-teacher distillation Miki et al. (2022). Thus, we learn policies end-to-end in a single stage, foregoing the need for multi-stage training pipelines (Hoeller et al., 2024). Rewards and policy training details can be found in section A.2. Camera latency can be large in real, so per-sensor latency randomization is used to improve Sim2Real (table 8).

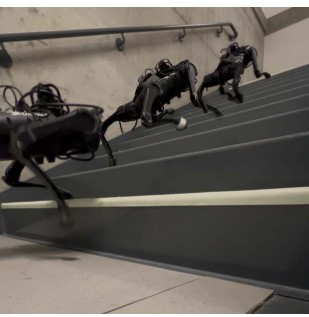

(b) Zero-shot deployment to real.

Figure 6: *Sim-to-real*: Gauss-Gym worlds enable training vision policies that transfer to real without fine-tuning.

### 4.2.1 NEURAL ARCHITECTURE

At the core of our framework is a recurrent encoder that fuses visual and proprioceptive streams over time. At each timestep, proprioceptive measurements are concatenated with DinoV2 (Oquab et al., 2023) embeddings extracted from the raw RGB frame. These combined features are passed through an LSTM, producing a compact latent representation that captures both temporal dynamics and visual semantics. The choice

of LSTM is motivated by the need for fast inference speed on the robot, thereby limiting the use of vanilla transformer architectures.

Two task-specific heads operate on this representation: *Voxel prediction head:* The latent vector is unflattened into a coarse 3D grid and processed by a 3D transposed convolutional network. Successive transposed convolution layers upscale this grid into a dense volumetric prediction of occupancy and terrain heights. In doing so, the shared latent representation has to capture the geometry of the scene. Visualized predictions are shown in fig. 7. *Policy head:* In parallel, a second LSTM consumes the latent representation together with its recurrent hidden state, and outputs the parameters of a Gaussian distribution over joint position offset actions. Additional training details, including observation spaces, scene configurations, and rewards, are provided in section A.2.

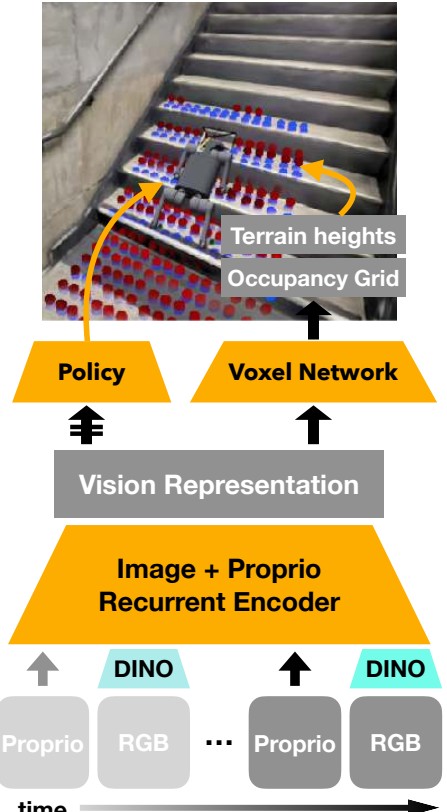

Figure 7: *Architecture for Visual Locomotion*: An LSTM encoder fuses proprioception with DinoV2 RGB features. Outputs feed into a 3D transpose conv head for occupancy and terrain prediction, and a policy LSTM that outputs Gaussian action distributions.

### 4.2.2 VISUAL LOCOMOTION RESULTS

While the task of stair climbing can be solved purely through geometric or blind locomotion Miki et al. (2022), it provides a valuable context for studying the behavior learned by our visual policy when approaching stairs. Our policy, trained on the Unitree A1 using RGB image inputs, learns to precisely place its feet on stairs and adapt its gait to avoid colliding with stair risers within the simulation, as illustrated in fig. 6a and Appendix fig. 11. Therefore, allowing the policy to robustly match commanded velocities across terrains. As a proof of concept, we successfully transfer this policy to the real world without additional fine-tuning (see fig. 6b). The policy and vision encoder are run as separate ROS nodes onboard the A1's Jetson TX2, with actions queried at 50Hz and image latents queried at 15Hz. Similarly, our policy, trained in simulation with a head-mounted camera on the Booster T1, learns to successfully navigate slopes.

### 4.2.3 VISUAL NAVIGATION RESULTS

The visual navigation tasks consist of a sparse goal tracking task in which the agent must navigate around obstacles to reach distant waypoints. To test the trained agent, we created an obstacle-field experiment (fig. 8). In this scenario a sparse goal was placed behind clutter, and a penalty region was introduced via a yellow patch on the floor. When the agent enters the penalty region it receives a negative reward signal during training. The RGB policy successfully avoided the patch, while the depth-only policy failed, demonstrating that RGB conveys rich semantic cues beyond geometric depth, enabling policies to reason about environmental semantics. Crucially, these results highlight the importance of using RGB input over depth-only sensing.

We furthermore performed a large-scale ablation of multiple design parameters. We tested our robots in 4 simulation scenarios (flat, steep, and short and tall stairs), as shown in Appendix table 2. In summary, not regressing on the voxel grid or not using a pre-trained DINO encoder reduces performance. Furthermore, training on a large number of scenes provides significant improvement in performance compared to using $10\%$ or $50\%$ of the scenes, highlighting the relevance of the seamless infrastructure to train across multiple scenes in GaussGym.

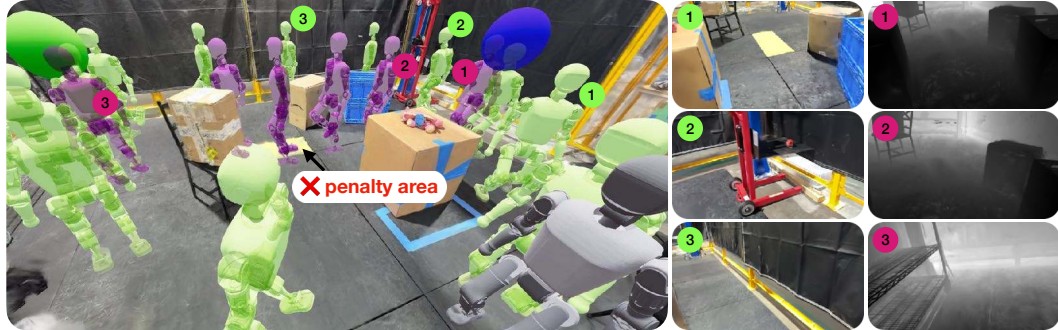

Figure 8: *Semantic reasoning from RGB*: In the sparse goal tracking task, the robot must cross an obstacle field where a yellow floor patch incurs penalties. The RGB-trained policy (green) perceives and avoids the patch, while the depth-only policy (purple) cannot detect it and walks through. This highlights how RGB provides semantic cues beyond geometric depth.

## 5 LIMITATIONS

Visual sim-to-real transfer remains a difficult and largely unsolved problem, and GaussGym offers a promising platform for developing algorithms to narrow this gap. In simulation, our vision-based policies learned to avoid high-cost regions and achieved precise foothold placement. Yet, further experiments are required to assess generalization across a broader set of tasks. For example, our walking policy was not evaluated on unseen staircases during training, and we observed a decline in the precise foot placement seen in simulation when transferring to real-world scenarios. Transferring visual policies to real hardware introduces additional challenges, including physical delays (e.g., image latency) and the reliance on egocentric observations. In contrast, geometry-based methods that leverage elevation maps and high-frequency state estimation (e.g., 400 Hz) substantially simplify the locomotion problem.

For tasks where visual information is critical—such as adhering to social norms (e.g., walking on a sidewalk or crosswalk)—GaussGym currently lacks automated mechanisms for generating cost or reward functions. Foundational language models could help shape agent behavior by defining these functions, but in this work we relied on hand-crafted cost terms.

Assets in GaussGym are initialized with uniform physical parameters (e.g., friction), which prevents accurate simulation of surfaces like ice, mud, or sand—limiting the connection between "how something looks and how it feels" Chen et al. (2024a). Additionally, adding support for other popular simulation backends such as IsaacSim Makoviychuk et al. (2021b) and MuJoCo Todorov et al. (2012) is under ongoing.

Although GaussGym builds on state-of-the-art vision models, it inherits their limitations. For example, Veo's outputs can be inconsistent, requiring re-prompting, and offer limited camera control. Future integration of more controllable and temporally consistent models, such as Genie 3 (DeepMind, 2025), presents a clear path to improvement. Additionally, a detailed analysis of how video-model-generated scenes impact policy generalization is required. Finally, our methods for generating worlds from video models cannot yet handle dynamic scenes or simulate fluids and deformable assets beyond the simple rigid-body physics provided by IsaacGym.

## 6 CONCLUSION

We present GaussGym, a fast, open-source photorealistic simulator for training visual locomotion and navigation policies directly from RGB. GaussGym supports scenes from real-world robot deployments, smartphone scans, video-generation models, and existing datasets. Policies trained in GaussGym exhibit vision-perceptive behavior in simulation and show partial transfer to real-world scenarios. With this work, we provide an open baseline for training visual navigation and locomotion policies to benefit the research community. Just as earlier generations of massively parallel, GPU-based physics simulators democratized geometric locomotion learning, we expect GaussGym to accelerate progress and spur new advances in vision-based locomotion and navigation.

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

## A APPENDIX

### A.1 ADDITIONAL SCENES AND MOTION BLUR

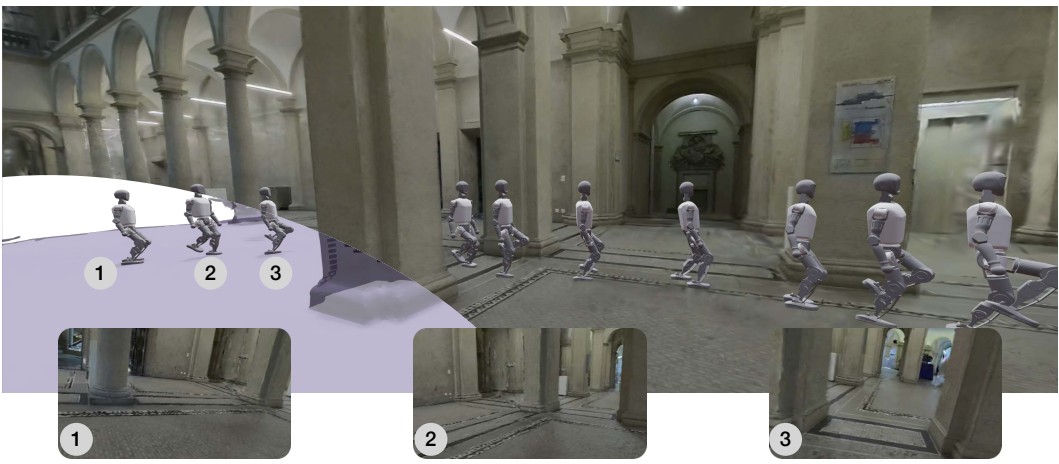

Figure 9: *Large photorealistic worlds*: GaussGym incorporates open-source datasets, such as Grand-Tour (Frey et al., 2025), which contains high quality scans of large areas. Shown above is a $20\,\text{m}^2$ GaussGym scene derived from GrandTour, including the mesh (purple) and robot POV renders.

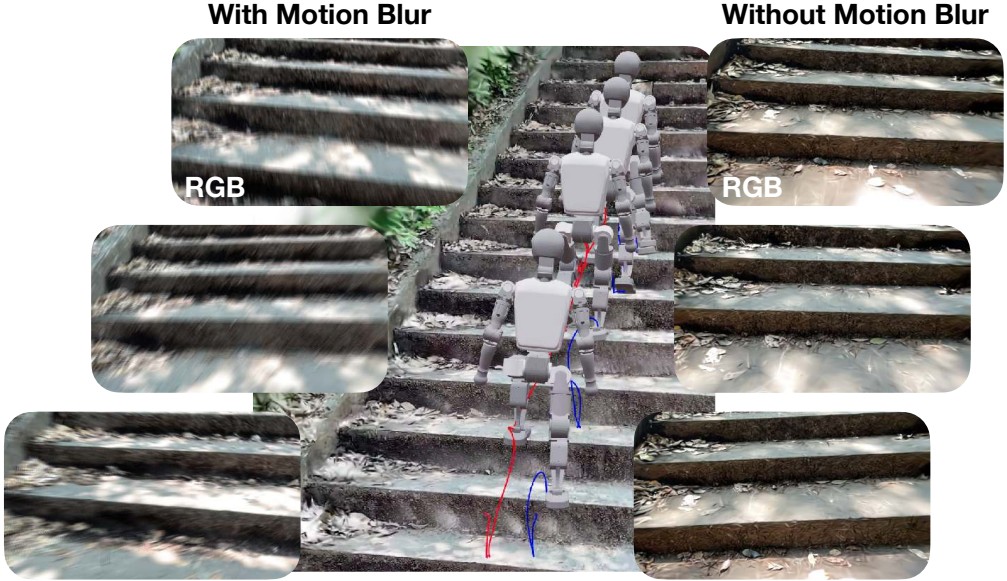

Figure 10: GaussGym proposes a simple yet novel to simulate motion blur. Given the shutter speed and camera velocity vector, GaussGym alpha blends various frames along the direction of motion. The effect is pronounced in jerky motions, for example when the foot comes into contact with stairs.

## A.2 POLICY LEARNING

| Scenario | Vision | | Blind | | Vision w/o voxel | | Vision w/o DINO | | Vision $\frac{1}{10}$ scenes | | Vision $\frac{1}{2}$ scenes | |
|---|---|---|---|---|---|---|---|---|---|---|---|---|
| | A1 | T1 | A1 | T1 | A1 | T1 | A1 | T1 | A1 | T1 | A1 | T1 |
| Flat | **100.0** | **100.0** | 98.1 | 97.2 | **100.0** | 98.3 | **100** | 96.7 | 94.3 | 99.2 | 99.0 | 99.2 |
| Steep | **99.3** | **97.1** | 89.4 | 87.6 | 91.9 | 87.0 | 95.6 | 91.5 | 88.1 | 88.3 | 95.5 | 94.1 |
| Stairs (short) | **98.7** | **97.4** | 80.8 | 72.3 | 85.2 | 82.7 | 92.3 | 87.5 | 79.7 | 74.8 | 86.3 | 84.9 |
| Stairs (tall) | **94.4** | **92.5** | 74.0 | 60.5 | 80.8 | 76.3 | 88.3 | 82.8 | 67.3 | 58.2 | 83.9 | 75.2 |

Table 2: Results for the **goal tracking** task. Each method has two subcolumns for robots A1 and T1. Bold numbers indicate the best performance per scenario.

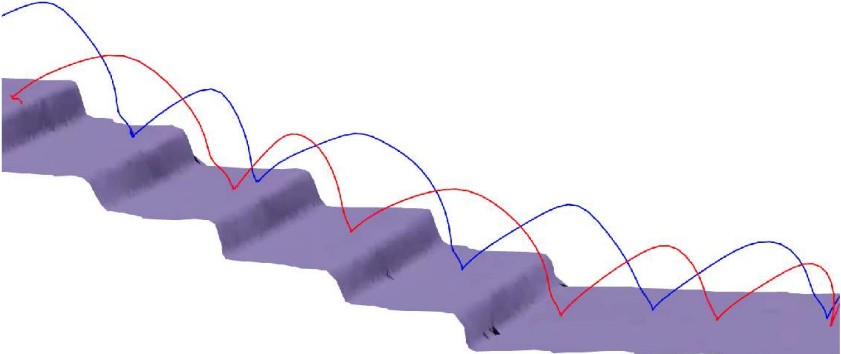

Figure 11: *A1 foot swing trajectory*: Foot trajectories for the visual locomotion policy in sim. The A1 learns to correctly place its front (red) and hind (blue) feet without stumbling on the stair edge. When approaching the stairs, A1 leads with the front foot, taking a large step to land securely in the middle of the second step, indicating that safe footholds can be directly inferred from vision.

| Reward | Expression | Weight |
|---|---|---|
| Ang Vel XY | $\|\omega\|^2$ | -0.2 |
| Orientation | $\|\alpha\|^2$ | -0.5 |
| Action Rate | $\|q_t^* - q_{t-1}^*\|^2$ | -1.0 |
| Pose Deviation | $\|q_t - \hat{q}\|^2$ | -0.5 |
| Feet Distance | $(f_{\text{left},xy} - f_{\text{right},xy}) < 0.1$ | -10.0 |
| Feet Phase | $1_{f,\text{contact}} \times \phi \leq 0.25$ | 5.0 |
| Stumble | $\|F_{f,xy}\| \geq 2\|F_{f,z}\|$ | -3.0 |

Table 3: General reward terms used for all tasks, their mathematical expressions, and associated weights used in training locomotion policies. $\omega$ is the angular velocity, $\alpha$ is the angle between the global up vector and the policy up vector, $q^*$ is the commanded action, $q$ is the current joint angle, $f$ is the foot position, $1_{f,\text{contact}}$ is the contact indicator function, $\phi$ is the current gait phase and $F$ is the foot contact force.

| Reward | Expression | Weight |
|---|---|---|
| Linear Velocity Tracking | $\exp(-\|v_{xy} - v_{xy}^*\|^2/0.25)$ | 1.0 |
| Angular Velocity Tracking | $\exp(-\|\omega_z - \omega_z^*\|^2/0.25)$ | 0.5 |

Table 4: Rewards used for the velocity tracking task. $v$ and $v^*$ are the current and desired base velocities. $\omega$ and $\omega^*$ are the current and desired yaw rates.

| Reward | Expression | Weight |
|---|---|---|
| Position tracking | $1_{t<1}(1 - 0.5\|r_{xy} - r_{xy}^*\|)$ | 10.0 |
| Yaw tracking | $1_{t<1}(1 - 0.5\|\psi - \psi^*\|)$ | 10.0 |

Table 5: Rewards used for the goal tracking task. We base our rewards on (Hoeller et al., 2024). $t$ is the remaining time to reach the goal. $r$ and $r^*$ are the current and desired base positions. $\psi$ and $\psi^*$ are the current and desired base yaws.

| Observation |
|---|
| Base Ang Vel $\omega_b$ |
| Projected Gravity Angle $\alpha$ |
| Joint Positions $q$ |
| Joint Velocities $\dot{q}$ |
| Swing phase $\phi$ |
| Image $I \in (640 \times 480)$ |

Table 6: Observations used across all tasks.

| Value | Range |
|---|---|
| Camera update rate | [8,20] Hz |
| Focal length | [290, 310] |
| Local camera position | [-0.02, 0.02] cm |
| Local camera orientation (degrees RPY) | [-1, 1] deg. |
| Base COM (XYZ) | [-10, 10] cm |
| Base Mass | [80, 120] % |
| Link COM | [-2, 2] cm |
| Motor strength | [90, 110] % |
| Joint stiffness | [90, 110] % |
| Joint damping | [90, 110] % |
| Friction | [0.3, 2.0] |
| Compliance | [0.5, 1.5] |
| Restitution | [0.1, 0.9] |
| Kick interval | [2.0, 5.0] s. |
| Kick velocity delta | [-1.5, 1.5] m/s |

Table 7: Domain randomization values.

| Observation | Range |
|---|---|
| Base Ang Vel $\omega_b$ | [0, 40] ms. |
| Projected Gravity Angle $\alpha$ | [0, 40] ms. |
| Joint Positions $q$ | [0, 40] ms. |
| Joint Velocities $\dot{q}$ | [0, 40] ms. |
| Swing phase $\phi$ | [0, 0] ms. |
| Image $I \in (640 \times 480)$ | [30, 200] ms. |

Table 8: Latency randomization values.

## A.3 ADDITIONAL COMPARISONS

### A.3.1 VR-ROBO

Here we compare the end-to-end policy and training architecture used in GaussGym to the hierarchical approach used in VR-Robo (Zhu et al., 2025). VR-Robo trains a high-level navigation controller which receives visual and proprioceptive information and outputs velocity commands. It also trains a low level controller which receives velocity commands and actuates the legged robot's motors to satisfy the high level command. Overall, we find that the GaussGym policy architecture / training approach performs significantly better; specifically in low data regimes. We hypothesize that this may be due to the high-level controller overfitting.

| Scenario | Vision | | Vision $\frac{1}{10}$ scenes | | Vision $\frac{1}{2}$ scenes | |
|---|---|---|---|---|---|---|
| | end-to-end | hierarchical | end-to-end | hierarchical | end-to-end | hierarchical |
| Flat | 100.0 | 100.0 | 94.3 | 86.5 | 99.0 | 95.7 |
| Steep | 99.3 | 98.1 | 88.1 | 75.7 | 95.5 | 82.4 |
| Stairs (short) | 98.7 | 98.0 | 79.7 | 54.7 | 86.3 | 75.5 |
| Stairs (tall) | 94.4 | 87.8 | 67.3 | 50.6 | 83.9 | 63.1 |

Table 9: Results for the **goal tracking** task. Each method has two subcolumns for end-to-end and hierarchical (VR-Robo style) controllers.

