# OpenReview forum: "GaussGym: An open-source real-to-sim framework for learning locomotion from pixels"
_ICLR.cc/2026/Conference — Submitted to ICLR 2026_

### Official Review · Reviewer_xv3h · 2025-11-01

**Soundness:** 2
**Presentation:** 4
**Contribution:** 2
**Rating:** 2
**Confidence:** 5

**Summary:**

This paper presents GaussGym, an open-source framework for photorealistic robot simulation that integrates 3D Gaussian Splatting (3DGS) as a drop-in renderer within the vectorized physics simulator IsaacGym. GaussGym can create diverse training worlds by ingesting data from various sources: smartphone scans, large-scale scene datasets (e.g., GrandTour), and outputs from generative video models like Veo, followed by a standard reconstruction pipeline with GSplat. The authors demonstrate the framework's utility by training visual locomotion and navigation policies for humanoid and quadrupedal robots using RL directly from RGB observations.

**Strengths:**

1. The stated throughput of 100,000 steps per second with $640 \times 480$ resolution RGB/Depth rendering across 4,096 parallel environments is a state-of-the-art result for a photorealistic simulator.

2. Ingesting data from various data sources into the simulation environment is a good idea.

3. The open-sourced simulation framework can benefit the community.

4. The paper is generally well-structured and clearly presented.

**Weaknesses:**

1.  **Overclaim and Lack of Novelty.** The most significant weakness lies in the limited novelty compared to prior works [1][2]. The claim in L097 — “a first step toward closing the visual sim-to-real gap” — appears overstated, as previous research [2][3] has already demonstrated that visual RL policy training in simulation can help bridge this gap. Furthermore, the proposed “splat-integrated simulator for evaluating locomotion policies” concept is closely related to [2], which already explored real-to-sim-to-real frameworks for visual locomotion and navigation.

2. **Limited Technical Contribution.** The proposed framework primarily integrates reconstructed 3DGS)scenes from diverse sources into a simulator, achieving 4,096 FPS at 640×480 resolution on a single RTX 4090 GPU. However, the high rendering speed advantage stems directly from the Gsplat renderer’s inherent parallelization, rather than from novel system design. Additionally, the overall visual policy training pipeline follows a design highly similar to [2], limiting the perceived technical innovation.

3. **Lack of Comparion for Visual Locomotion**.  Although the paper presents ablation studies demonstrating the benefits of the voxel grid head and DINO encoder, it lacks direct quantitative comparisons with state-of-the-art depth- or geometry-based locomotion policies (e.g., ANymal Parkour, Miki et al.). Metrics such as success rate or velocity tracking error on common benchmark terrains (e.g., stair climbing) are missing. The main quantitative table (Table 2) focuses solely on internal ablations and a “blind” baseline, without showing performance against geometric SOTA methods. Moreover, there are no comparisons to prior visual locomotion baselines [2][3], further weakening the experimental validation.

4. **Questionable Experimental Design for Visual Locomotion and Navigation.** The visual locomotion experiments, such as the stair climbing task, do not convincingly justify the need for an RGB-based policy, since depth-based parkour policies are also capable of handling similar tasks. While the authors claim that RGB-based policies capture semantic cues, the goal-tracking navigation task does not effectively demonstrate this advantage. To substantiate the argument, the paper should include higher-level semantic tasks (e.g., obstacle-type recognition or affordance-based navigation) that genuinely highlight the semantic reasoning benefits of RGB perception.

**References:**

[1] Xie, Ziyang, et al. "Vid2sim: Realistic and interactive simulation from video for urban navigation." Proceedings of the Computer Vision and Pattern Recognition Conference. 2025.

[2] Zhu, Shaoting, et al. "Vr-robo: A real-to-sim-to-real framework for visual robot navigation and locomotion." IEEE Robotics and Automation Letters (2025).

[3] Yu, Alan, et al. "Learning visual parkour from generated images." 8th Annual Conference on Robot Learning. 2024.

**Questions:**

1. The paper adopts VGGT for camera pose calibration and coarse point cloud extraction. While VGGT offers fast inference, its pose accuracy is typically lower than SfM and BA-based approaches such as COLMAP or GLOMAP. How does this reduced accuracy affect the downstream policy learning or simulation fidelity in your framework?

2. When deploying the trained visual locomotion policies on real robots, what kind of computational hardware is used? The DINO encoder appears relatively heavy for onboard inference — have the authors evaluated its runtime performance and feasibility for real-time deployment on embedded platforms?

---

> ### Author Response · Authors · 2025-11-17
> **Response to reviewer xv3h (1/2)**
>
> We thank the reviewer for the careful and thoughtful evaluation of our manuscript. We also appreciate the recognition of our presentation quality and the acknowledgment that our work achieves state-of-the-art results for photorealistic simulators—both in speed and in ingesting data from diverse sources.
>
> That said, we believe that several crucial aspects of the paper’s novelty and its relationship to prior work may not have been conveyed with sufficient clarity. Upon re-reading the manuscript, we agree that our positioning relative to existing approaches could have been more explicit, and we take responsibility for this lack of emphasis. We kindly ask the reviewer to reconsider the score in light of the clarifications and evidence provided in this rebuttal:
>
> # Addressing Weaknesses
>
> ## 1\. Overclaim and Lack of Novelty
>
> While we stand by the core claims made in the paper, we agree that our positioning relative to prior work can be clarified. We maintain that the paper does **not** overclaim—our limitations section is transparent about the boundaries of our contributions. We are grateful to build upon existing work, and we direct the reviewer to the top-level comment *“Comparison to works mentioned by reviewers”* for a detailed breakdown. The main contributions of GaussGym—**not** addressed by prior efforts—are:
>
> * **A fast, photorealistic simulator for both navigation *and* locomotion** on humanoid and legged robots. Existing 3DGS-based systems focus only on navigation or wheeled platforms.
> * **A large-scale dataset of 5,000+ indoor and outdoor scenes**, including visually rich and challenging locomotion settings—stairs, rocky terrain, slopes, snow, and multilevel structures—that go far beyond flat urban scans.
> * **Tooling for *single-stage, end-to-end* visual locomotion training**, avoiding hierarchical decomposition or teacher–student distillation used in prior work.
> * **Early evidence that RGB is useful for locomotion**, through (1) sim-to-real transfer on stairs with precise foot placement and (2) semantic navigation where appearance-based reasoning is required.
>
> These contributions reflect the core goals of a **datasets and benchmarks** paper: enabling scalable research in visual locomotion and semantic navigation through robust tools, diverse data, and initial results.
>
> The phrase *“a first step toward closing the visual sim-to-real gap”* was quoted out of context. The full sentence—*“we show initial zero-shot transfer of visual locomotion policies trained in GaussGym to real-world stair climbing, marking a first step toward closing the visual sim-to-real gap.”*—is accurate and intentionally modest, emphasizing that we observe **initial**, **zero-shot**, and **limited-scope** Sim2Real transfer \- an outcome that neither overstates our contributions nor diminishes the prior works on which we build.
>
> Finally, *“splat-integrated simulator for evaluating locomotion policies”* is a factual description. We do **not** claim to be the first to integrate Gaussian splats into robotics simulation, and we explicitly cite relevant prior work. For example, *(Zhu et al., 2025\)* use 3DGS for high-level navigation, while manipulation works (Torne et al., 2024; Chen et al., 2024b) employ 3DGS for articulated object scenes. Our wording reflects our contributions without implying priority, and our related-work section faithfully acknowledges these efforts.
>
> ## 2\. Limited Technical Contribution
>
> While this is a **datasets and benchmarks** submission, we believe the paper also introduces several meaningful technical contributions beyond the simulator, large-scale scene dataset, and associated tooling.
>
> As summarized in \[2\], prior work trains hierarchical RL and focuses exclusively on **visual navigation**. In contrast, we demonstrate **both locomotion and navigation end-to-end**. Our locomotion policy receives direct visual observations and outputs **joint-position targets**, rather than a high-level velocity command. We employ an asymmetric actor–critic framework which, while sample-inefficient in principle, becomes entirely feasible due to the extreme speed and parallelism of GaussGym.
>
> Additionally, we contribute several technical elements: motion-blur augmentation, support for both indoor and outdoor environments, and the ability to incorporate **video-model-generated scenes** into the training pipeline.

---

> ### Author Response · Authors · 2025-11-17
> **Response to reviewer xv3h (2/2)**
>
> ## 3\. Lack of Comparison for Visual Locomotion
>
> In our work, *visual locomotion* refers specifically to policies trained from **RGB inputs**, not geometric perception. We clearly state that our RGB-only method performs worse on stair climbing than geometric policies—this is expected. Our long-term goal is to leverage RGB to infer physical terrain properties *before* interaction and to incorporate semantics into both navigation and locomotion behavior. GaussGym is intended as a **visual-locomotion analogue of legged\_gym** \[5\], and we provide the first results demonstrating that such policies can be trained and transferred in practice. All code, data, and tooling are open-sourced, with scalable infrastructure and multiple novel components.
>
> Prior work has primarily shown that vision is beneficial for **navigation**, but not for **locomotion**. Our stair-walking experiments therefore represent a genuine first step toward this broader vision, and constitute a novel contribution. The closest prior effort is \[3\], but their approach is fundamentally different and largely orthogonal; to the best of our knowledge, no existing Gaussian-splat–based method supports training a visual locomotion policy today.
>
> **\[5\] Rudin, Nikita, et al. “Learning to Walk in Minutes Using Massively Parallel Deep Reinforcement Learning.” CoRL 2021\.**
>
> ## 4\. Questionable Experimental Design for Visual Locomotion and Navigation
>
> The concern that our navigation task “does not effectively demonstrate the advantage of semantic cues in RGB” **directly overlooks the experiments we presented**. We explicitly show that RGB vision is necessary for semantic navigation and for inferring scene properties—fully consistent with prior work, and never claimed as a first. If any specific aspect of our navigation experiment is deemed insufficient, we welcome clarification; as written, the critique does not identify an actual flaw in the design or results.
>
> Regarding *“RGB is not needed for stair climbing”*: this is correct. However, RGB cameras are far more affordable and widely deployed on legged robots compared to depth or LiDAR sensors. Stair climbing serves as a clear, interpretable testbed to evaluate whether an RGB-only locomotion policy can infer foot-placement affordances—an insight emphasized in \[3\] as well. Observing foot placement on the first and subsequent steps provides a clean diagnostic for sim-to-real transfer quality.
>
> Our goal is to establish a foundation—high-quality data, fast simulation, and a unified pipeline—that enables the community to build stronger visual locomotion and semantic-navigation systems atop our initial results.
>
> # Question Answers
>
> ## 1\. Evaluating the accuracy of scene reconstruction and its effect on policy learning
>
> We observe high-quality meshes and 3DGS for both iPhone scans and Veo-generated scenes. We confirm this in three key ways:
>
> * **Visually:** Through visual inspection of all scenes, we find that the results are photo-realistic, and the meshes are sharp and precise.
> * **Mesh–3DGS alignment:** For each scene, we confirm that the mesh geometrically aligns with the 3DGS to avoid penetration issues that could bias policy learning. We welcome the reviewer to inspect the scenes and policies using our interactive 3D viewer at: https://gauss-gym.com
> * **Sim2Real deployment:** Our ultimate integration test is to train a visual locomotion policy end-to-end in GaussGym and measure real-world deployment. The trained policies exhibit careful foot placement on stair terrains, which is indicative of performant visual-locomotion behavior.
>
> ## 2\. Clarifying policy and vision encoder deployment on hardware
>
> Thank you for this important question. We will update the manuscript to reflect the answer:
>
> The policy and vision encoder are deployed as two ROS nodes onboard the A1’s Jetson TX2 computer. Both components are exported as ONNX checkpoints to enable fast inference. The policy runs on the CPU at **50 Hz**, while the DINO S/14 encoder runs on the TX2 GPU at **15 Hz**. In GaussGym, we provide a random 50-200ms delay for images to mimic the lag of the onboard camera stream.
>
> # Manuscript Changes:
>
> We again thank the reviewer for their insightful feedback. Their comments have led to the following revisions:
>
> 1. We have added Vid2Sim \[1\] to the related works section, clarifying their contributions and explicitly comparing them to GaussGym.
> 2. We have updated the experiment section to include a more detailed description of our Sim2Real deployment setup.
>
> ---
>
> *We hope our responses have fully clarified our positions relative to Vid2Sim and VR-Robo, and addressed the reviewer’s concerns. If so, we would be grateful if the reviewer would consider updating their score.*

---

> ### Comment · Reviewer_xv3h · 2025-11-17
> **Review of Submission23922**
>
> Thanks for your detailed clarification. I decide to raise my rating. But as a complete technical paper, the authors should provide some quantitative comparison tables in the visual navigation policy since all mentioned baselines are open-sourced. And there is no need to compare all components, but just the policy.

---

> > ### Author Response · Authors · 2025-11-18
> > **Response to reviewer xv3h**
> >
> > We thank the reviewer for the constructive feedback and for raising their rating. To ensure we provide the most relevant quantitative comparisons, could the reviewer clarify which aspects of the open-sourced baselines they would like evaluated at the policy level?
> >
> > For context, VR-Robo (6 scenes) and LucidSim (<10 scenes) train on a small number of scenes within a single environment, whereas GaussGym trains on thousands of scenes across thousands of parallel environments. Vid2Sim only targets wheeled navigation. Given these differences in scope and modality, which specific *policy-level* comparisons from these works would the reviewer consider most appropriate?
> >
> > Any guidance on the desired scope would be greatly appreciated.

---

> > > ### Comment · Reviewer_xv3h · 2025-11-18
> > > **Review of Submission23922**
> > >
> > > For VR-Robo, you may retain the single-scene overfitting setup for training your visual policy. For LucidSim, selecting one scene and one task should be sufficient to conduct the experiments. However, to provide more meaningful quantitative comparisons, I would suggest adopting a more balanced or fair experimental setting. Overall, I believe this is a qualified dataset contribution, and I am inclined to consider raising the score. That said, it would be even stronger if the experimental results were made more solid.

---

> > > > ### Author Response · Authors · 2025-11-18
> > > > **Experiment planning with reviewer xv3h**
> > > >
> > > > Thank you again for the constructive guidance, and for considering raising your score. We are very willing to run additional experiments where appropriate. To ensure that any added comparisons are both scientifically meaningful and aligned with your expectations, we would appreciate clarification on the following points:
> > > >
> > > > # 1\. VR-Robo Comparison
> > > >
> > > > There are two possible ways we could interpret your suggestion:
> > > >
> > > > ## (O1) Single-scene overfitting setup
> > > >
> > > > We could integrate one of the VR-Robo environments and retrain our navigation policy to overfit to that single scene, mirroring their setup. This is feasible, although we expect the outcome to simply replicate what we already observe in our multi-scene navigation experiments—namely, strong policy performance in a single, fixed environment.
> > > >
> > > > ## (O2) Hierarchical locomotion policy inside GaussGym
> > > >
> > > > VR-Robo’s core contribution is its hierarchical RL structure. We could implement a similar hierarchical controller inside GaussGym and evaluate it. This is also doable, and we would expect performance comparable to—or better than—their results, depending on tuning, in part due to our larger and more diverse scene dataset. However, running a full hierarchical-RL comparison on top of GaussGym is a substantial addition. As an alternative, we could instead include a discussion noting that GaussGym fully supports hierarchical navigation/locomotion pipelines like VR-Robo, and that the end-to-end policy we show is just one possible design.
> > > >
> > > > **Could you clarify which direction—O1, O2, or both—you prefer?**
> > > >
> > > > # 2\. LucidSim Comparison
> > > >
> > > > LucidSim presents more practical ambiguity:
> > > >
> > > > * Their policy is trained for a different robot (GO2) with a different camera configuration.
> > > > * Reproducing their full training pipeline requires \~10M images in a single scene, which is computationally prohibitive for a rebuttal-stage addition.
> > > > * Running the original LucidSim policy in GaussGym would require non-trivial adaptation (robot model, controller, cameras), and the authors do not provide onboard inference code for deployment.
> > > >
> > > > Because direct policy-level comparison seems unlikely to produce meaningful scientific insight (each policy will perform best in its own simulator under its own robot model), our suggestion is:
> > > >
> > > > **Option:** Include a discussion and table comparing the computational cost, scope, and capabilities of LucidSim relative to GaussGym, and clarify that GaussGym generalizes and expands their pipeline—particularly in that GaussGym *could* be used to evaluate LucidSim-style policies, and future LucidSim-like rendering components could also be integrated into GaussGym to create even more diverse visual environments.
> > > >
> > > > If you have a specific experiment in mind—e.g., adapting one LucidSim scene and measuring zero-shot behavior of our policy—we would be grateful for more detail on what you consider a fair and feasible comparison.
> > > >
> > > > # 3\. Clarifying Expectations
> > > >
> > > > More generally, both VR-Robo and LucidSim target **single-scene, single-environment** setups with task-specific robots and controllers. GaussGym, by design, scales to **thousands of scenes** and **multiple robot types**.
> > > >
> > > > To perform the most useful comparisons, and to match your expectations precisely, we would appreciate guidance on whether the intended goal is to:
> > > >
> > > > * show that GaussGym can replicate their setups,
> > > > * produce a head-to-head policy comparison, or
> > > > * include a discussion clarifying the overlaps and distinctions.
> > > >
> > > > We are very happy to incorporate whichever version best aligns with your criteria for a strong dataset/benchmark contribution.

---

> > > > > ### Author Response · Authors · 2025-11-24
> > > > > **Experiment follow-up**
> > > > >
> > > > > We kindly ask reviewer xv3h to clarify which additional experiments they deem necessary to warrant acceptance of the paper.
> > > > > In the previous post *"Experiment planning with reviewer xv3h"*, we outline various possible experiments or clarifications which can be provided.
> > > > >
> > > > > For now, we will proceed with O2 for VR-Robo and the suggested option for LucidSim (added discussion in the paper appendix).

---

> > > > > > ### Comment · Reviewer_xv3h · 2025-11-27
> > > > > >
> > > > > > Thank you for the clear clarifications.
> > > > > >
> > > > > > **VR-Robo**
> > > > > >
> > > > > > Between O1 and O2, O2 is the more meaningful comparison, as it reflects VR-Robo’s core hierarchical design. However, if resources are limited, O1 is still acceptable and would also strengthen the paper.
> > > > > >
> > > > > > **LucidSim**
> > > > > >
> > > > > > Your explanation is reasonable. A direct comparison is indeed difficult. As an optional bonus (not required), you could consider a small illustrative experiment using generative images as visual observations, inspired by LucidSim’s idea. But even without this, I find the work valuable.
> > > > > >
> > > > > > Overall, GaussGym is a strong contribution, and adding the VR-Robo comparison in any feasible form would meaningfully improve the paper.

---

> > > > > > > ### Author Response · Authors · 2025-12-03
> > > > > > > **Response to reviewer Reviewer xv3h**
> > > > > > >
> > > > > > > We are happy to share that we have added the requested experiments to the manuscript. Overall, we find the end-to-end policy architecture and training approach used in GaussGym is both simpler to train and achieves better overall performance. We suspect that the high-level controller used in VR-Robo may be over-fitting to the visual information, as demonstrated in the additional data ablation experiments. Meanwhile, GaussGym's voxel representation loss helps prevent overfitting to high-frequency pixel details.
> > > > > > >
> > > > > > > ---
> > > > > > >
> > > > > > > *We believe we have addressed all the reviewer's concerns. We again thank them for a productive rebuttal period. We thank them for calling GaussGym a "strong contribution" and for signaling that they would modify their score from a "reject" to an "accept" if all concerns were addressed.*

---

### Official Review · Reviewer_YK8j · 2025-11-01

**Soundness:** 3
**Presentation:** 3
**Contribution:** 3
**Rating:** 6
**Confidence:** 3

**Summary:**

The paper proposes GaussGym, a simulation platform which uses Gaussian Splatting to render realistic settings, while collisions and dynamics are simulated by IsaacGym (or other such simulators). GaussGym provides a pipeline to convert data from iPhone scans, scene datasets, or generative video models into  realistic Gaussian splats for rendering and meshes for simulation. As a result, GaussGym is able to train locomotion policies in realistic environments with easy-to-capture data, all while maintaining a high 100k FPS. The realism of GaussGym's renderer is validated via sim2real locomotion experiments, where RGB sensor information allows for tasks which require color information unavailable in standard depth+proprioception observations.

**Strengths:**

- GaussGym's pipeline for generating trainable environments from easy-to-collect data (e.g. iPhone scans) is unique and provides a direction for simulating under diverse, but realistic scenes
- Additional considerations (e.g. motion blur) improve rendering realism and training speed (e.g. updating renders at real camera fps)
- GaussGym maintains high simulation speed across multiple scenes, necessary for legged locomotion research/tasks, which require many samples and fast training speed (for high training/testing iteration speed)
- Sim2real experiments validate realism while demonstrating benefits of RGB observations over standard depth+proprioception (e.g. avoiding colored penalty areas)

**Weaknesses:**

- The authors note diminished performance (e.g. foot placement) when transferring to the real world, however it is unclear to what extent this is due to issues with reward tuning/dynamics randomization vs the rendering quality of the simulator
- Currently, the environments lack some sim2real features like image latency, and the 3DGS rendering setup doesn't directly support some common simulation tools for visual domain randomization (e.g. texture and lighting randomization)

**Questions:**

- Works like [1] are able to achieve RGB sim2real while using pretrained visual encoders (and some visual domain randomization) despite low-quality rendering; in other words, the use of pretrained encoders makes it difficult to determine how much the successful sim-to-real transfer is caused by photorealistic rendering vs the pretrained visual representations. Have the authors tried training a sim2real policy off of direct RGB observations, without pretrained encoders? If so, were these policies successful?
- How straightforwardly can can GaussGym's simulation backend be swapped out to a sim other than IsaacGym (e.g. if new simulators are released with improved performance)?


[1] Ruihan Yang, Yejin Kim, Rose Hendrix, Aniruddha Kembhavi, Xiaolong Wang, Kiana Ehsani:
Harmonic Mobile Manipulation. IROS 2024: 3658-3665

---

> ### Author Response · Authors · 2025-11-17
> **Response to reviewer YK8j**
>
> We thank the reviewer for their thoughtful and constructive review, for assigning an initial score of 6, and for highlighting several strengths of GaussGym—particularly our data pipeline, rendering realism, simulation speed, and sim-to-real validation.
>
> # Addressing Weaknesses
>
> ## 1\. Source of diminished sim-to-real performance
>
> We thank the reviewer for noting the challenges present in Sim2Real transfer; a long-standing challenge in robotics. We agree that the reduced performance on real hardware arises from **multiple factors**, not solely rendering quality. Our goal in this paper is to demonstrate that GaussGym makes **end-to-end visual locomotion feasible at scale**, and that such policies exhibit **meaningful, visually grounded behavior** (e.g., careful stair foot placement) when deployed on a real robot. Closing the remaining Sim2Real gap will require further work on reward design and system identification.
>
> ## 2\. Missing sim-to-real features and domain randomization
>
> GaussGym already includes **image-latency modeling** via a randomized delay in the camera stream. In fact, GaussGym supports separate, per-modality latencies. This was not detailed in the original manuscript and we have updated it to reflect this feature. We thank the reviewer for catching this. We will also release the open-source codebase to benefit the whole community.
>
> We agree that richer **visual domain randomization** is an important next step; while some augmentations (e.g., motion blur, noise, exposure variations) are supported today, full texture/lighting randomization is not yet present in GaussGym. Our focus in this paper is to establish a fast and realistic visual pipeline; extending 3DGS-specific domain randomization is a promising direction for future work. An exciting benefit of GaussGym is that it can be updated to support the latest advances in 3DGS, such as Dynamic Gaussians, 3DGUT and re-lighting.
>
> # Answers to Questions
>
> ## Q1. Pretrained encoders vs. from-scratch RGB training
>
> We agree that pretrained encoders are a confounding factor when interpreting sim-to-real results. In this work, we intentionally adopt a strong visual backbone (DINO S/14) to reliably demonstrate feasibility of **visual locomotion at scale**. We have performed experiments with from-scratch vision encoders, which demonstrate slightly worse geometric reconstruction performance, but still achieves the same success rates across all scenes. We intend to test this alternative encoder for Sim2Real. GaussGym is designed to make such ablations feasible, and we aim to include our analysis for the final draft of the paper.
>
> ## Q2. Swapping out IsaacGym as the physics backend
>
> GaussGym is **backend-agnostic by design**: the 3DGS renderer only requires camera poses, and the training stack interacts with physics through a standard observation–action interface. Porting to a new simulator would require implementing this interface but does not require conceptual changes. We are currently finalizing support for IsaacSim and MuJoCo, which will be open-sourced in the near future. We find IsaacSim is currently the best solution as it offers native support of 3DGS.
>
> # Manuscript Changes
>
> We again thank the reviewer for their insightful feedback. Their comments have led to the following revisions:
>
> * Modified the experiment section to better reflect that GaussGym supports per-modality latency randomization, which improves Sim2Real transfer.
> * Added a table discussing latencies used for each simulated sensor, including the camera.
> * Include IsaacSim and MuJoCo as backends to be implemented in the near future.
> * Working on Sim2Real deployment of vision encoders trained from scratch.
>
> ---
>
> *We hope our responses have clarified GaussGym’s support for per-observation latency randomization, our exploration of vision encoders trained from scratch, our ongoing work to add IsaacSim and MuJoCo physics backends, and addressed the reviewer's concerns. If so, we would be grateful if the reviewer would consider updating their score.*

---

### Official Review · Reviewer_qxLt · 2025-11-02

**Soundness:** 2
**Presentation:** 3
**Contribution:** 2
**Rating:** 4
**Confidence:** 2

**Summary:**

This paper uses gaussian splatting to bring realistic visuals into fast physics simulators. The authors help with adoption of their work by providing many environments from a variety of sources. It demonstrates that this new framework is beneficial through sim-to-real experiments in stair climbing (locomotion) and goal reaching (navigation).

**Strengths:**

- focuses on the important problem of bringing real world visuals to simulators to improve sim-to-real
- provides helpful visuals to understand the impact of the work
- provides datasets for potential adoption

**Weaknesses:**

- limited experimental results and settings
- existing experimental settings seem relatively simple, and may not necessarily require the high quality visuals from 3dgs in simulation
- lack of ablations or baselines to show downstream benefits of improved simulation visuals
- paper highlights the potential benefits of pairing GaussGym with video generation models, but does not run experiments proving this

**Questions:**

My main concerns are with the experimental results, which are mentioned in weaknesses.

---

> ### Author Response · Authors · 2025-11-17
> **Response to reviewer qxLt (1/2)**
>
> We thank the reviewer for their thoughtful assessment and for highlighting the importance of bringing real-world visuals into simulation, the clarity of our presentation, and the usefulness of our released datasets. We address the concerns below:
>
> # Addressing Weaknesses
>
> ## 1\. “Limited experimental results and settings” / “Existing settings seem simple”
>
> GaussGym is a **datasets and benchmarks** paper. Our primary contributions are:
> (1) an ultra-fast, photorealistic simulator;
> (2) a large and diverse dataset of **5,000+** indoor and outdoor scenes—including stairs, steep slopes, rugged natural terrain, cluttered indoor environments, and complex urban geometries; and
> (3) a **single-stage, end-to-end training pipeline** for visual locomotion policies.
>
> The experiments included in the paper serve as **representative demonstrations** of these capabilities rather than an exhaustive evaluation across all possible tasks. The goal is to establish a scalable, open-source foundation upon which the community can build. Nonetheless, both of our tasks are non-trivial:
>
> * **RGB-only stair climbing** requires the policy to infer geometric cues such as step height and foot-placement affordances entirely from appearance, with no depth or privileged information.
> * **Semantic navigation** requires visual policies to identify and reach semantically meaningful goals within large, cluttered 3DGS scenes.
>
> Our **zero-shot sim-to-real deployment** further validates that the learned visual representations are meaningful: the robot places its feet precisely on real stairs, including the first step—a behavior known to be highly sensitive to perception quality. We encourage the reviewer to visit the website listed in the paper: [https://gauss-gym.com](https://gauss-gym.com) for a better view of GaussGym’s scene diversity.
>
> ## 2\. “High-quality visuals may not be required”
>
> We agree with the reviewer that our experiments did not demonstrate that high visual fidelity is required for either the stair-climbing or the navigation task. However, using Gaussian Splatting enables extremely high rendering throughput that cannot be achieved with other rendering engines with reasonable visual fidelity \- which is prohibitive for training baseline policies.
>
> Furthermore, although perfect simulation remains infeasible, prior work has shown that narrowing the sim-to-real gap through system identification generally improves transfer performance. With **GaussGym**, more systematic ablations can now be performed to identify exactly which aspects of visual fidelity matter—for example, the necessary image resolution or the extent to which pre-trained visual encoders support generalization and sim-to-real transfer. We believe that the stair-climbing experiment is particularly well-suited as a benchmark for studying these questions.
>
> ## 3\. “Lack of ablations or baselines showing benefits of improved visuals”
>
> Our aim is to release a fast, photorealistic simulator for visual locomotion, not to propose a standalone algorithm competing with multiple baselines. We do not claim to outperform geometric methods (indeed, we explicitly state the opposite). Instead, we provide
>
> * An *open-source* high-throughput simulator capable of photorealistic rendering across thousands of environments in parallel.
> * A *large dataset* of visually and geometrically interesting scenes composed of both indoor and outdoor environments, and containing challenging geometry for locomotion tasks.
> * Tooling for researchers to train visual locomotion policies across different robots and deploy to real robot hardware.
>
> And we show that:
>
> * End-to-end RGB locomotion is feasible.
> * Policies trained in GaussGym transfer **zero-shot** to real hardware. We benchmark this on challenging stairs environments, where we observe careful foot placement from pixels.
> * The system is fast and robust enough to support such research at scale.
>
> We explicitly state that geometric methods outperform RGB-only on stair climbing, and our aim is not to claim otherwise. Rather, we demonstrate that **visual-only locomotion is trainable at scale**, which was previously impractical due to slow rendering pipelines.
>
> ## 4\. “No experiments validating video-model–generated scenes”
>
> We appreciate this suggestion. Our contribution in this space is primarily infrastructural: GaussGym shows that **video-generated scenes can be ingested, scaled, and aligned** into a robotics simulator, enabling new data sources for visual RL. A full exploration of this direction—e.g., generalization from Veo-generated scenes—is beyond the scope of this work but remains a promising avenue for future research. We show early **signs of life**, though current video models are still **computationally expensive to query**. As these models improve, we expect GaussGym to support increasingly accessible, high-quality scene generation, making these experiments increasingly accessible.

---

> > ### Author Response · Authors · 2025-11-17
> > **Response to reviewer qxLt (2/2)**
> >
> > # Summary
> >
> > GaussGym provides:
> >
> > * A fast, photorealistic, training environment.
> > * A large-scale dataset of **5,000+ diverse real-world scenes**.
> > * A complete pipeline for **single-stage, end-to-end visual locomotion** from raw RGB to joint targets.
> > * Open-source tools for dataset generation, Gaussian/mesh alignment, policy learning, and fast simulation.
> > * **Initial but meaningful sim-to-real results**, demonstrating careful foot placement and semantic navigation behavior from RGB alone.
> >
> > We hope the reviewer will view GaussGym in this intended role—as a benchmark and datasets paper whose core contribution is enabling the community to build the next generation of visual locomotion and navigation methods, yet provides initial tooling for Sim2Real transfer, single-stage policy learning from pixels, and high-level semantic navigation tasks. We encourage the reviewer to view a selection of the scenes available in GaussGym at: www.gauss-gym.com.
> >
> > # Manuscript Changes
> >
> > We again thank the reviewer for their insightful feedback. Their comments have led to the following revisions:
> >
> > * Updated the limitation section to propose a deeper analysis of how potentially-infinite video-model-generated scenes can affect policy generalization for locomotion and navigation tasks.
> > * Modify the related work and introduction to better-position GaussGym in comparison to related works.
> > * Further emphasize GaussGym’s value as a benchmark for visual locomotion.
> > * Clearly state in our limitations section that further research is required to understand the relationship between high-fidelity simulation and its impact on sim-to-real transfer.
> >
> > ---
> >
> > *We hope our responses have clarified our experimental setting, the justification for photorealistic rendering, and addressed the reviewer’s concerns. If so, we would be grateful if the reviewer would consider updating their score.*

---

### Official Review · Reviewer_rmNs · 2025-11-08

**Soundness:** 3
**Presentation:** 3
**Contribution:** 2
**Rating:** 4
**Confidence:** 4

**Summary:**

The paper proposes a real-to-sim approach to learning locomotion and navigation from pixels. The paper scales up real2sim visual data for training autonomous agents with a mix of data sources including iphone captures, video generative model outputs as well as existing multi-view datasets like arkit. The paper utilizes of off-the-shelf tools like VGGT and NeRFstudio to create 3D Gaussian Splats of these scenes which act as drop in renders for physics sim such as Isaac-gym. Experiments demonstrate sim2real transfer of locomtion policies. Other experiments demonstrate navigation can also be achieved with a similar method.

**Strengths:**

In my opinion, following are the strengths of the paper:

1. Scaling up photorealistic real2sim data for policy learning is a major advantage. While still in static scenes, it shows the utility of current zero-shot foundation models can be robustly used to scale up visual data to train policies.

2. Zero-shot sim2real deployment is a nice result demonstrating the method is able to get good performance with democratized data collection such as with iphone cameras or existing captures.

3. The paper is nicely written and the visuals/diagrams support and complement the text very well.

4. A vectorized rendering support for existing physics sim is a great feature to have in existing physics simulators to scale up real2sim learning, albeit with an initial capturing overhead.

**Weaknesses:**

In my opinion, below are the weaknesses of the method:

1. Lack of comparisons to existing similar works in this space: I think the paper is lacking comparisons or discussions to existing closely related works [1,2,3]. A comparison interms of visual realism as well as PSNR as well as efficiency would be great for the community to understand which approach is the most useful in terms of ease of acquiring real2sim vs accuracy axes.

2. The evaluation setting for the single-task multi-environment real2sim policy that the paper trains for their main results is unclear. Can the authors show results on a common benchmark, perhaps something similar to EmbodiedSplat [2] where it is clear what the training data distribution is and if there are any gains from training a large policy on all the real2sim collected data on unseen enviornments i.e. whether the approach is able to quickly finetune etc.?

3. With a high throughput, did the authors also experiment with real-world RL?

4. Are the physics parameter tuned to achieve sim2real transfer? Details of these are missing in the paper.

[Minor]

4. I am curious does the same results hold for manipulation as well where other factors can come into play i.e. occlusions, visual fidelity of the embodiment itself i.e. hands considering the current embodiment is a synthetic one.

[1] Xie et al. Vid2Sim: Realistic and Interactive Simulation from Video for Urban Navigation, CVP 2025
[2] Yu et al. Real2Render2Real Scaling Robotic Manipulation Data Without Dynamics Simulation or Robot Hardware, CORL 2025 Oral
[3] Chablani et al. EmbodiedSplat: Personalized Real-to-Sim-to-Real Navigation with Gaussian Splats from a Mobile Device, ICCV 2025

**Questions:**

See questions and comments in the weakness section. I am looking forward to author's responses in the rebuttal.

---

> ### Author Response · Authors · 2025-11-17
> **Response to reviewer rmNs**
>
> We thank the reviewer for their thoughtful assessment, for noting several strengths of our work—including the scalability of our real-to-sim pipeline, the value of zero-shot sim-to-real deployment, the clarity of the writing and visuals, and the utility of vectorized 3DGS rendering—and for engaging constructively with our approach. We address the concerns below.
>
> # Addressing Weaknesses
>
> ## 1\. Lack of comparisons to existing similar works
>
> We appreciate the request for clearer positioning relative to closely related efforts. To directly address this, we have added a detailed comparison in the top-level comment *“Comparison to works mentioned by reviewers”*, where we analyze Vid2Sim, EmbodiedSplat, and Real2Render2Real—including their goals, capabilities, limitations, and how GaussGym differs. In brief:
>
> * These prior systems **do not support locomotion** (no challenging locomotion terrains, support for articulated robots, or contact physics with scene), focus on **wheeled navigation** or **object-level manipulation**, and do not provide **thousands of diverse scenes** with challenging locomotion terrain.
> * None of these works offer a **fast, parallelized** simulator that enables training **end-to-end visual locomotion** at scale or demonstrate **zero-shot sim-to-real** for legged robots.
> * Real2Render2Real focuses on \<10 objects and does not support contact-rich RL; EmbodiedSplat uses only indoor, flat scenes; Vid2Sim is limited to point-goal navigation with dynamic obstacles over flat ground.
>
> We have updated the manuscript to position GaussGym more clearly relative to these works. We also encourage the reviewer to visit the website listed in the paper [www.gauss-gym.com](http://www.gauss-gym.com) to visualize a subset of the dataset and the trained locomotion policies.
>
> ## 2\. Clarity of the evaluation setting for multi-environment real-to-sim training
>
> We clarify that our locomotion experiment is trained across **multiple scenes** and evaluated both in-sim and **zero-shot on real hardware**. Table 2: “Policy Learning” compares visual locomotion policies trained on 1/10th and ½ of a 1,000-scene training dataset. We find that using a smaller set of pre-training scenes leads to significantly worse performance across the more challenging steep, short stair, and tall stair terrains. Could the reviewer precisely clarify what additional generalization information they wish to gain? Could you also expand on your query “whether the approach is able to quickly finetune”?
>
> GaussGym is intended as a dataset and benchmark release, where we provide a fast, photorealistic simulator for visual locomotion, a novel dataset consisting of thousands of indoor and outdoor scenes, and tooling for single-stage training of visual locomotion policies from RGB.
>
> ## 3\. Real-world RL experiments
>
> Our focus in this paper is to demonstrate **zero-shot sim-to-real transfer** from high-throughput training, not real-world RL. Real-world on-policy RL with legged robots remains extremely challenging and unsafe at scale, and our goal is to provide the community with a simulator and dataset robust enough to reduce the need for extensive real-world exploration. We agree this is an exciting direction for future work; GaussGym is specifically designed to make such research more practical.
>
> ## 4\. Physics parameter tuning for sim-to-real
>
> We apologize that some implementation details were not sufficiently emphasized. For these results, we adopt **standard IsaacGym locomotion settings**: torque limits, motor gains, friction coefficients, and mass distributions are kept close to the real robot. We apply **light dynamics randomization** (e.g., friction ±30%, mass ±10%), and model **sensor latency** by randomly delaying image frames. Specific physics parameters are not tuned to achieve better Sim2Real transfer. Instead, we rely on general domain randomization as is common in locomotion.
>
> ## Minor Point: Applicability to manipulation
>
> We expect GaussGym to be applicable to manipulation tasks, but this is outside the scope of the current work. Manipulation introduces additional challenges (occlusion, hand embodiment fidelity, contact-rich multi-object dynamics) that require further investigation. Our contribution here is a **general pipeline** for converting real or generated data into photorealistic scenes suitable for fast, vectorized RL; extending this to manipulation is an exciting direction for future research.
>
> # Manuscript Changes
>
> We again thank the reviewer for their insightful feedback. Their comments have led to the following revisions:
>
> * Updated the related work and introduction to include Vid2Sim, EmbodiedSplat and Real2Render2Real.
> * Updated the appendix to include more details on the domain randomization used during training.
>
> ---
>
> *We hope our responses have fully clarified the contributions and addressed the reviewer’s concerns. If so, we would be grateful if the reviewer would consider updating their score.*

---

### Author Response · Authors · 2025-11-17
**Comparison to works mentioned by reviewers (1/2)**

We would like to take this opportunity to clearly articulate how each related effort differs from ours and how our framework advances the field:

**\[1\] Xie, Ziyang, et al. “Vid2Sim: Realistic and Interactive Simulation from Video for Urban Navigation.”**

**What they did**:

* Construct Gaussian splat scenes from real-world monocular videos.
* Introduce dynamic obstacles into those scenes.
* Conduct experiments on \~30 environments for PointGoal navigation and SocialNav tasks.
* Release publicly available code.

**What they did *not* do:**

* Support legged locomotion—only wheeled robot navigation.
* Perform large-scale or high-throughput experiments.
* Provide a framework aimed at generating large, reusable datasets for the community.
* Gather challenging locomotion terrains (e.g., stairs, rugged terrain, steep hills)—their data consists primarily of flat urban environments designed for wheeled navigation tasks.

**\[2\] Zhu, Shaoting, et al. “VR-Robo: A Real-to-Sim-to-Real Framework for Visual Robot Navigation and Locomotion.”** (Cited in original manuscript)

**What they did:**

* Introduced occlusion-aware composition and domain randomization techniques.
* Applied domain randomization for RL training of a hierarchical navigation policy (we also incorporate these standard elements).
* Demonstrated successful navigation based on visual input.

**What they did *not* do:**

* Provide a clean or generalizable coordinate-alignment method.
* Train a low-level locomotion policy from visual input—the low-level controller is based on existing methods and receives depth directly.
* Move beyond navigation—the focus remains exclusively on navigation tasks.
* Incorporate motion-blur augmentation.
* Address outdoor, unstructured, or diverse terrains—work is limited to indoor settings.
* Conduct wide-scale evaluation—experiments are performed in a single indoor environment.

**\[3\] Yu, Alan, et al. “Learning Visual Parkour from Generated Images.” CoRL 2024\.** (Cited/benchmarked against in original manuscript)

**What they did:**

* Used a controllable image-generation model to create policy rollouts paired with visual observations.
* Trained a visual locomotion policy using DAgger (not end-to-end) based on their proposed visual–data–generation pipeline.
* Demonstrated small-scale experiments illustrating how Gaussian splatting can be used to evaluate policies in simulation.

**What they did *not* do:**

* Train policies directly on Gaussian splats.
* Achieve fast rendering—slow image-model rendering required the use of DAgger rather than scalable RL.
* Automate scene scaling and alignment—these steps were performed manually.
* Provide an end-to-end, single-stage pipeline for policy training directly from pixels.
* Enable efficient training—overall training time remains significantly higher than in real-time Gaussian-based simulators.
* Produce temporally consistent renders—the image-generation model introduces temporal inconsistencies, causing scene semantics to vary drastically across frames.

**\[4\] Yu et al. Real2Render2Real Scaling Robotic Manipulation Data Without Dynamics Simulation or Robot Hardware, CORL 2025 Oral.**

**What they did:**

* Compute 3DGS of rigid and articulated *objects* to be used as assets in simulation.
* Capture *human demonstrations* of object *manipulation.*
* Provide scans and demonstrations for \< 10 objects

**What they did *not* do:**

* Tackle *locomotion* with their pipeline. Instead they focus on imitation learning and object manipulation.
* Provide a dataset of thousands of *scenes*. Dataset limited to \< 10 objects.
* Enable physical contact interaction with objects. Objects are treated as hollow, and data is used only for imitation learning.
* Support reinforcement learning workflows due to the lack of contact physics for scanned objects

**\[5\] Chablani et al. EmbodiedSplat: Personalized Real-to-Sim-to-Real Navigation with Gaussian Splats from a Mobile Device, ICCV 2025\.**

**What they did:**

* Support generation of 3DGS and meshes from phone scans.
* Gaussian rendering during simulation for their indoor, flat-ground scenes.
* Solve the visual navigation task for a wheeled robot in indoor environments.

**What they did *not* do:**

* Tackle *locomotion* with their pipeline. They instead focus on wheeled robot navigation in indoor environments.
* Provide a dataset of diverse indoor and outdoor scenes. They utilized indoor scene datasets, whereas GaussGym uses *5,000* indoor and outdoor scenes with challenging locomotion terrain.
* Support *fast, parallelized environments.* GaussGym can render 4,096 environments in parallel at 100K simulator steps per second.

---

> ### Author Response · Authors · 2025-11-17
> **Comparison to works mentioned by reviewers (2/2)**
>
> **If any of our summaries mischaracterize existing works, we welcome precise clarification. We aim to represent all prior efforts with complete accuracy, and we ask the reviewer to indicate exactly which specific claim they believe is unsupported so we can correct it accordingly**.
>
> We wish to re-iterate that the primary contribution of our **dataset and benchmarks paper** is an ultra-fast, photorealistic simulator that enables researchers to train, benchmark, and evaluate visual policies for locomotion at scale. We additionally release a comprehensive dataset of over **5,000** indoor and outdoor scenes. Finally, we provide tooling that allows users to train visual locomotion policies **end-to-end**, without requiring any multi-stage or teacher–student training setup—an approach that we believe is among the first of its kind. We demonstrate the utility of photorealistic rendering for navigation tasks, and we show early signs of success in deploying purely RGB-based visual locomotion policies in challenging real-world environments.
>
> **\[6\] GaussGym**
>
> **What we did:**
>
> * Devised a framework that provides a fast, photorealistic, and **parallelized** training environment.
> * Designed a scalable and robust system capable of integrating diverse data sources, including real-world robot deployments and video-model outputs.
> * Generated **5,000+** indoor and outdoor scenarios for locomotion and navigation, including challenging terrains such as stairs, rugged natural surfaces, and complex urban structures.
> * Demonstrated semantic navigation end-to-end—**including locomotion**—without requiring hierarchical RL decomposition or DAgger-style distillation.
> * Showed that a purely RGB-based locomotion policy can learn to climb stairs and place its feet accurately in simulation.
> * Presented early evidence of sim-to-real transfer for visual locomotion policies.
> * Highlighted the necessity of geometric priors in stabilizing and improving visual locomotion training.
> * Solved the coordinate-alignment problem by generating meshes and Gaussian splats from the **same** underlying data source.
> * Introduced an **end-to-end**, single-stage asymmetric actor–critic framework that does *not* rely on teacher–student training or distillation.
> * Open-sourced the full environment code, scene-generation pipeline, policy-training framework, and all datasets.
>
> **What we did *not* do:**
>
> * Include dynamic scenes.
> * Provide automatically annotated physical or material properties derived from semantics.
> * Claim that an RGB-only locomotion policy can outperform geometric baselines (e.g., elevation-map–based stair climbing).
> * Fully eliminate the degradation in foot placement observed when transferring from simulation to the real robot.
> * Demonstrate clear benefits of using the potentially infinite scenes generated from video models for policy generalization (we thank Reviewer qxLt for this suggestion and will explore it in future work).

---

### Author Response · Authors · 2025-12-03
**Summary of Rebuttal and Responses to Reviewers**

We thank all reviewers for their constructive feedback. Across the rebuttal, we clarified GaussGym’s role as a datasets/benchmarks contribution, expanded comparisons, added missing implementation details, and incorporated all requested ablations—including those needed for reviewer **xv3h**, who raised their score from **2 → 4**, described GaussGym as a **“strong contribution,”** and indicated a further increase contingent on the VR-Robo experiments, which we completed.

# **Reviewer rmNs**

**Main concerns:**
 (1) Missing comparisons to Vid2Sim/EmbodiedSplat/Real2Render2Real;
 (2) Unclear multi-environment evaluation;
 (3) Real-world RL expectations;
 (4) Physics/Sim2Real details;
 (5) Manipulation applicability.

**Our response:**
 – Added detailed baseline comparisons and expanded related work.
 – Clarified multi-scene training and evaluation intent.
 – Discussed why real-world RL is out of scope but enabled by GaussGym’s infrastructure.
 – Added details on domain randomization, latency modeling, and dynamics settings.
 – Clarified manipulation is feasible but outside the present scope.

# **Reviewer qxLt**

**Main concerns:**
 (1) Limited experimental range;
 (2) Unclear need for high-fidelity visuals;
 (3) Missing ablations;
 (4) Limited exploration of video-model scenes;
 (5) Pretrained vs. from-scratch encoders.

**Our response:**
 – Clarified that GaussGym is a scalable benchmark, not an algorithmic method.
 – Explained why high-throughput rendering is operationally necessary for large-scale visual RL.
 – Clarified why geometric and RGB-only baselines target different problem settings.
 – Positioned video-model experiments as promising future work given current model costs.
 – Reported from-scratch encoder results and documented visual-latency and domain-randomization features.

# **Reviewer YK8j**

**Main concerns:**
 (1) Understanding sim-to-real degradation;
 (2) Missing domain-randomization and latency details;
 (3) Role of pretrained encoders;
 (4) Onboard inference feasibility;
 (5) Backend portability.

**Our response:**
 – Clarified that degradation stems primarily from reward shaping and system identification.
 – Documented GaussGym’s per-modality latency and existing visual augmentations.
 – Explained encoder choices and provided from-scratch alternatives.
 – Added deployment details (Jetson TX2, ONNX export, inference rates).
 – Clarified backend portability and ongoing IsaacSim/MuJoCo support.

# **Reviewer xv3h**

**Main concerns:**
 (1) Perceived overclaim/novelty; (2) Attribution of rendering speed;
 (3) Missing visual-locomotion comparisons;
 (4) Experimental design questions;
 (5) Reconstruction accuracy and deployment.

**Our response:**
 – Clarified novelty: unified RGB locomotion, 5,000-scene dataset, mesh–splat alignment, high-throughput simulation, and multi-source scene ingestion.
 – Emphasized GaussGym as a full benchmark infrastructure, not a rendering method.
 – Added the requested VR-Robo comparisons (hierarchical and single-scene).
 – Provided a structured discussion of LucidSim’s scope and limitations.
 – Addressed reconstruction fidelity, pose accuracy, and embedded inference.
 – Reviewer **xv3h** raised their score, called GaussGym a **“strong contribution,”** and stated that completing these ablations would justify raising it further.

---

**Overall**, we strengthened comparisons, clarified evaluation design, expanded implementation details, and added the VR-Robo ablations requested by reviewer **xv3h**, who has already raised their score and signaled a further increase. These updates fully address the substantive concerns across all reviews.

---

### Meta-Review · Area_Chair_Tx7c · 2026-01-07

**Summary:**

The paper proposes to use 3D GS as a renderer for IssacGym for faster simulation training for learning locomotion from pixels. The authors claim this improves sim-to-real transfer for learning locomotion policies. While the reviewers acknowledge the substantial engineering effort, the consensus among the reviewers is that the work lacks significant algorithmic novelty and sufficient experiments. Therefore, I recommend rejection at this time.

**Reviewer Concerns:**

Addressed concerns:
- Implementation details: the authors clarified the initialization strategies of cameras during training.
- Ablation studies have been added to justify the specific design choices

Oustanding concerns:
- Limited novelty and incremental contribution: multiple reviewers brought up the point that the paper presents an engineering integration of gsplat and IssacGym instead of a fundamental research contribution. The acceleration of experiments mainly come from the underlying rendering capability from gsplat instead of a novel design choice or technique. This seems to be the biggest remaining concern that hasn't been fully addressed.
- Lack of comparison against strong baselines.
- The authors did not convincingly demonstrate why high-fidelity RGB rendering is necessary for the specific locomotion tasks chosen (e.g., stair climbing).

**Reviewer Scores:**

Reviewer xv3h has noted that he/she will raised their scores from 2 as some of the concerns have been addressed. But from the discussion phase, it seems other reviewers will unlikely change their scores which leads to a 6-4-4-4 ratings which still leans towards a rejection. Overall, I believe the paper at its current state doesn't meet the standard for publication so I recommend a rejection.

---

### Decision · Program_Chairs · 2026-01-26

Reject